# Evaluating the specificity of flavivirus proteases in *Aedes aegypti* cells for dengue virus 2-derived cleavage sites

**Alexius O. Dingle[1,2], Zach N. Adelman[1] ***

**1** Department of Entomology, Texas A&M AgriLife, Texas A&M University, College Station, TX, United States of America, **2** Interdisciplinary Graduate Program in Genetics and Genomics, Texas A&M University, College Station, TX, United States of America

* zachadel@tamu.edu

**Data Availability Statement:** All relevant data are within the manuscript and its Supporting Information files.

## Abstract

Flaviviruses are a diverse group of RNA viruses known for their significant impact on human health worldwide. We generated a series of reporters that included cleavage sequences from the dengue virus type 2 polyprotein and co-transfected with plasmids encoding various flavivirus proteases into *Aedes aegypti* cells, followed by fluorescent imaging and western blot analysis for the determination of proteolytic cleavage. Recombinant flavivirus NS2B3 proteases from medically significant and insect-specific flaviviruses were able to process reporters encoding cleavage sequences from the dengue virus type 2 polyprotein *in vitro* including proteases from dengue virus types 1–4, Zika virus, yellow fever virus, *Aedes* flavivirus, and cell-fusing agent virus. Reporters were not cleaved when transfected cells were infected with dengue virus type 2. Endoplasmic reticulum tethered reporters were also cleaved by protease alone but not by infectious virus. These results shed light on the ability of multiple flavivirus proteases to cleave sequences derived from outside of their genome and raise new questions concerning the requirements for effective cleavage by flavivirus proteases *in trans*.

## Introduction

Flaviviruses are single-stranded, positive-sense RNA viruses that belong to the *Flaviviridae* family. The flavivirus genome encodes three structural proteins [capsid (C), membrane precursor (prM), and envelope (E)] and seven nonstructural proteins (NS1, NS2A, NS2B, NS3, NS4A, NS4B, and NS5). The viral genome is translated into a single polyprotein and processed by cellular and virally encoded proteases [1]. The flavivirus protease consists of NS3, which contains an N-terminal serine protease domain, and the NS2B co-factor, hence NS2B3 [2–4]. Post-proteolytic processing, the non-structural proteins form replication complexes that produce nascent RNA for new virions [2, 5].

Many flaviviruses are arthropod-borne (arboviruses) and are transmitted to humans through infected mosquito or tick bites [6]. Flaviviruses that are of significant human health concern include Zika virus, yellow fever virus (YFV), and dengue virus (DENV). Between

**Funding:** This project was supported by the National Institute of Allergy and Infectious Diseases (NIAID) of the National Institutes of Health (NIH) under award number R01AI149608. The funder had no role in the design of this study. There was no additional external funding received for this study.

**Competing interests:** The authors have declared that no competing interests exist.

2015–2017, there were more than 200,000 confirmed cases of ZIKV reported in the Americas [7, 8]. In the case of YFV, in 2018 there were approximately 109,000 severe cases of yellow fever and 51,000 deaths in Africa and South America [9]. DENV stands as the most prevalent and geographically widespread arbovirus, with an estimated 96 million cases manifesting clinically out of the approximately 390 million infections that occur annually worldwide [10, 11]. Currently, vaccines exist for DENV (Dengvaxia), YFV (YF-Vax), Japanese encephalitis virus (JEV) (IXIARO, SA 14-14-2, etc.), and tick-borne encephalitis virus (TBEV) (Ticovac and Encepur), but there are often issues with supply and demand or limitations as to who is eligible for vaccination, as in the case of DENV [12–15]. There are also no antiviral drugs for flaviviruses available, so treatment options are limited to palliative care [16]. For these reasons, most efforts to prevent virus transmission focus on controlling their vectors, with the primary one being *Aedes aegypti*. Chemical insecticides have generally served as the main mechanism for controlling this insect, and long-term use has burgeoned insecticide resistant *Ae. aegypti* populations [17]. Many insecticides are also detrimental to the environment and harmful to nontarget species [18, 19]. These problems necessitate the development of innovative vector control methods.

In response to the shortcomings of traditional vector control strategies, much research has gone into the development of different genetic-based pest management strategies. Genetic-based pest management primarily falls under two modalities: population suppression and population replacement [20–22]. Population suppression aims to reduce the numbers of a target population by introducing transgenes that result in lethality or sterility. SIT (Sterile Insect Technique) relies on the release of large numbers of radiation or chemo-sterilized male mosquitoes to mate with wild females, resulting in no or fewer viable offspring [23]. RIDL (Release of Insects Carrying a Dominant Lethal) is essentially a modified version of SIT that uses a heritable, late-acting lethal gene that results in the death of mosquito progeny prior to reaching functional adulthood [24]. Both SIT and RIDL approaches face challenges; while SIT demands ongoing releases and high costs in addition to the often reduced fitness of sterilized males, RIDL has historically relied on antibiotic administration for suppressing expression of the lethal gene, which raises concerns about ecological impacts and additional cost and logistical complexities [23, 25–28]. On the other hand, population replacement strategies have the benefit of not eradicating the vector species, and thus prevent any potential environmental ramifications that could result from a species' complete elimination [29].

Some previously reported approaches in *Ae. aegypti* include RNAi and antibody based methods that reduced vector competence for DENV [30–34]. Additionally, a transgenic *Ae. aegypti* strain was developed that expressed a DENV2 activatable antagonist of the Inhibitor of Apoptosis (IAP) [35]. In this case, DENV2 induced apoptosis in mosquito cells that were actively infected with the virus and increased mortality in DENV2 infected transgenic mosquitoes when compared to infected wild-type mosquitoes [35]. As for ZIKV, virus transmission has been blocked in transgenic mosquitoes expressing a polycistronic cluster of small synthetic RNAs [36]. A more recent approach generated transgenic mosquitoes that become semi-paralyzed followed by subsequent death upon ZIKV infection [37]. Though effective, these methods are all specific to individual flaviviruses or at most all DENV serotypes. An ideal system would render *Ae. aegypti* unable to transmit all flaviviruses, and there is much value in the development of such a mechanism.

In addition to researching ways to prevent flavivirus transmission by vectors, efforts have also been dedicated to developing tools that can be used for diagnostics, identifying therapeutic targets, and better understanding flavivirus infection kinetics. A cell-based fluorescent reporter was able to detect and monitor NS2B3 activity during flavivirus infection in live cells [38]. Flavivirus protease activity was detected by the resultant GFP fluorescence increase in cells that

expressed fluorescently-quenched NS2B3 cleavable reporters [38]. Another study developed a dual-fluorescent reporter system to monitor flavivirus infection and endoplasmic reticulum (ER) manipulation [39]. NS2B3 cleavage of the reporter led to the translocation of one fluorescent protein into the nucleus, whereas before cleavage, it remained colocalized with another fluorescent protein at the ER [39]. Both of these studies have in common the ability of their cell-based reporters to detect infection by DENV2, ZIKV, and YFV via NS2B3 protease activity. This is not very surprising, as flavivirus cleavage sequences have been shown to be fairly conserved in prime side residues important for proteolysis [4, 40]. These findings, although derived from studies in mammalian cells, can inform the development of transgenic *Ae. aegypti* strains with NS2B3-activated effectors to prevent the transmission of multiple flaviviruses.

In order to develop transgenic mosquito strains that exhibit conditional mortality specific upon infection with any flavivirus, we sought to investigate the extent to which flavivirus proteases could process cleavage sequences derived from DENV2 as well as a synthetic consensus sequence. This was achieved by adapting the previously described fluorescent activatable reporter of NS2B3 protease activity to assays in *Ae. aegypti* cells [38]. We generated a series of EGFP-based reporters that were co-transfected with plasmids encoding various flavivirus proteases into *Ae. aegypti* cells, followed by fluorescent imaging and western blot analysis for the determination of proteolytic cleavage. We found that recombinant flavivirus NS2B3 proteases from medically significant and insect-specific flaviviruses were able to process reporters encoding cleavage sequences from the DENV2 polyprotein *in vitro*, though cleavage was not observed upon virus infection.

## Materials and methods

### Synthesis of reporter constructs and molecular cloning

All reporter plasmids (quenched and tethered) and viral protease expression vectors were commercially synthesized (Epoch Life Science, Missouri City, TX). The DNA sequence for both NS2B and the serine protease domain of NS3 were included for the viral proteases as previously described [41]. EGFP-RRRRSAG was removed via Q5 Site-Directed Mutagenesis (New England Biolabs, Ipswich, MA) from another previously synthesized reporter construct, pSLfa-PUb-HA-EGFP-RRRRSAG-mCh-Sec61γ-T2A-NS2B-NS3(DENV2), to create the effector, CDENV2. Site-directed mutagenesis was also used to generate an effector encoding a nonfunctional version of the protease by replacing the catalytic serine of NS3 with alanine (CDENV2-S135A). The remaining viral proteases were digested from their parent vectors and subcloned into the Pst1 and Sph1 restriction sites of CDENV2. All constructs were sequence-confirmed (Eton Bioscience, San Diego, CA) and purified using a midiprep kit for endotoxin-free plasmid DNA (NucleoBond Xtra Midi EF, Machery-Nagel, Düren, Germany).

### Cell culture

*Ae. aegypti* A20 cells were maintained in Leibovitz's L-15 (L15) medium (Invitrogen, Carlsbad, CA) supplemented with 10% FBS (Atlanta Biologicals, Minneapolis, MN), 2% tryptose phosphate broth (Invitrogen, Carlsbad, CA) and 1% penicillin/streptomycin (hereafter "complete" L15 = cL15) at 28°C in plug-seal T75 flasks.

### Cell transfection

A20 cells at 80% confluency were detached by scraping in 10 ml of fresh cL15, passed through a 10-mL serological pipette several times to dissociate clumps, and counted using the trypan

blue viability method. For cell lysate collection, $3–3.5 \times 10^5$ cells were seeded in the wells of 6-well plates with cL15 added to a final volume of 2 mL, followed by incubation overnight at 28˚C. The following day, the medium was replaced with 1.8 mL of fresh cL15. For transfection, 250 µL of Opti-MEM™ (Invitrogen, Carlsbad, CA) was mixed with 2.5 µL of reporter plasmid (1 µg/µl) or 1.25 µl each of the reporter and effector, and 5 µL of TransIT®-Insect Transfection Reagent (Mirus Bio LLC, Madison, WI). Transfection mixes were vortexed, centrifuged, incubated at room temperature for 15 minutes, and then added dropwise to the seeded cells. Transfected cells were incubated at 28˚C for 96 hours before harvesting cell lysate as described below. For imaging non-infected cells, $3 \times 10^4$ cells were seeded in Ibidi 8-well µ-slides coated with ibiTreat (Ibidi, Fitchburg, WI) and cL15 added to a final volume of 300 µL. The cells were incubated overnight at 28˚C and the medium was replaced the following day with 270 µL of fresh cL15. For transfection, 45 µL of Opti-MEM™ (Invitrogen, Carlsbad, CA) was mixed with 0.5 µL of reporter plasmid (1 µg/µl) or 0.25 µl of the reporter and effector, and 1 µL of TransIT®-Insect Transfection Reagent (Mirus Bio LLC, Madison, WI). The mixes were vortexed, centrifuged, and incubated at room temperature for 15 minutes. A volume of 28 µl of each transfection mix was added to the seeded cells. The cells were then incubated at 28˚C for 96 hours before imaging. For imaging infected cells and their controls, $1 \times 10^6$ cells were seeded in plug-seal T25 flasks, followed by incubation overnight at 28˚C. The following day, the medium was replaced with 5 mL of fresh cL15. For transfection, 1.9 mL of Opti-MEM™ (Invitrogen, Carlsbad, CA) was mixed with 19 µL of reporter plasmid (1 µg/µl), and 38 µL of TransIT®-Insect Transfection Reagent (Mirus Bio LLC, Madison, WI). The mixes were vortexed, centrifuged, and incubated at room temperature for 15 minutes. A volume of 648 µl of transfection mix was added to the seeded cells.

## Viruses and infection of cells

DENV2 New Guinea C isolate and ZIKV strain Mex-81 were obtained from the Kevin Myles lab (Texas A&M University, College Station, TX). A20 cells were left uninfected, or infected 24 hours post-transfection with DENV2 or ZIKV at an MOI of 0.01 and were allowed to adsorb for 1 hour with constant rocking. Cells were incubated at 28˚C for 144 hours, followed by imaging or harvesting of cell lysates and analysis by western blot as described below.

## Imaging

A20 cells were imaged using a Zeiss Axio Observer microscope using a 40x objective (Carl Zeiss Microscopy, White Plains, NY). A20 cells expressing the reporters alone as well as the reporters and effectors were imaged 96 hours post-transfection with exposure times of 150 ms for EGFP, 700 ms for DsRed, and 9 V 10 ms for Brightfield. Transfected cells infected with ZIKV or DENV2 and their controls were imaged 144 hours post-infection with exposure times of 58 ms for EGFP and 9 V 10 ms for Brightfield.

## Intracellular crosslinking

Transfected A20 cells expressing reporters alone or co-expressing reporters and viral proteases were scraped from the wells of 6-well plates and deposited into 2 ml microcentrifuge tubes. The cells were pelleted by centrifugation at $500 \times g$ for 5 minutes and washed three times with ice-cold 1X PBS (pH 8.0) before resuspension in a final volume of 90 µL. A 25 mM stock solution of disuccinimidyl suberate (DSS) (Thermo Scientific Pierce, Waltham, MA) was prepared by dissolving 2 mg of DSS in 216 µL of dry DMSO. Stock solution was added to the cells to a final concentration of 2.5 mM. After the cells were incubated at room temperature for 30 minutes, the reaction was quenched by adding 1 M Tris to a final concentration of 20 mM.

Following incubation of the quenching reaction for 15 minutes at room temperature, the cells were pelleted by centrifugation at 500 x *g* for 5 minutes, and the lysates isolated and analyzed by western blot as described below.

## Protein isolation, SDS-PAGE, and western blotting

Transfected cells were scraped from wells and the cell suspension deposited into 2 ml micro-centrifuge tubes. The cells were pelleted by centrifugation at $500 \times g$ for 5 minutes, after which the supernatant was decanted. The cells were washed with 1X PBS, centrifuged at $500 \times g$ for 5 minutes, and the supernatant decanted. Cells were lysed in RIPA buffer [50 mM HEPES pH 7.4, 150 mM sodium chloride, 1.0% NP-40, 0.5% sodium deoxycholate, 0.1% SDS, 0.1 M DL-dithiothreitol, and 1X protease inhibitor cocktail (Cell Signaling Technology, Danvers, MA)]. The lysates were rotated end-over-end at 4˚C for 30 minutes, centrifuged at maximum speed (~$18,000 \times g$) for 20 minutes, and transferred to fresh tubes with the insoluble portions discarded. Protein concentrations were measured using the Quant-iT™ protein assay kit and Qubit fluorometer (Invitrogen, Carlsbad, CA). Equivalent total protein was used for the preparation of samples that were run on an SDS-PAGE gel and transferred to nitrocellulose membrane (BioRad, Hercules, CA). Membranes were stained with Ponceau S to verify transfer, blocked with PBS-T + 4% non-fat milk for one hour, and blotted with a 1:1000 dilution of polyclonal anti-GFP (A01388, GenScript, Piscataway, NJ), rabbit polyclonal anti-DENV2 NS2B (GTX124246, GeneTex, Irvine, CA), or rabbit polyclonal anti-ZIKV NS2B (GTX133308, GeneTex, Irvine, CA) as primary antibodies. Polyclonal HRP-conjugated goat anti-rabbit (A00098, GenScript, Piscataway, NJ) was used as a secondary antibody at a dilution of 1:10,000. Membranes were subjected to 3 x 5-minute washes with PBS-T after antibody incubations. Bands were visualized using the ECL™ Prime Western Blotting Detection Reagent (Cytiva, Marlborough, MA) and images were recorded by autoexposure using the iBright™ 1500 (Invitrogen, Carlsbad, CA).

## Image analyses and statistical tests

Image analyses were performed with ImageJ (Fiji) software (National Institutes of Health, Bethesda, MD). Mann-Whitney or independent t-tests and Kruskal-Wallis tests were performed using Prism 9 software (GraphPad, La Jolla, CA); values of $p < 0.01$ and $p < 0.05$ were considered significant, respectively.

# Results

## Validation of quenched-EGFP reporters and DENV2-NS2B3 expressing effector in *Ae. aegypti* A20 cells

We used previous sequence alignments of DENV and ZIKV protease cleavage sites to identify the corresponding eight polyprotein cleavage sites of other medically significant flaviviruses transmitted by *Ae. aegypti* [40]. Based on our generated alignment, the DENV2 NS3, NS3/4A, and NS4A cleavage sites were the most conserved of the eight, thus we predicted that these sites would be cleaved by the greatest number of exogenous flavivirus proteases (Fig 1). Our quenched-EGFP reporter design was based off of previously described cell-based reporters used in mammalian cells [38, 42, 43]. The reporter consisted of the EGFP ORF, a linker encoding one of the NS2B3 cleavage sites, and a C-terminally fused quenching peptide (Fig 2). The quenching peptide (QP) suppresses the fluorescence of EGFP by preventing formation of the β-barrel, which is necessary for chromophore maturation [43]. Our expectation was that the viral proteases would cleave the linkers encoding the cleavage sites, ultimately leading to a

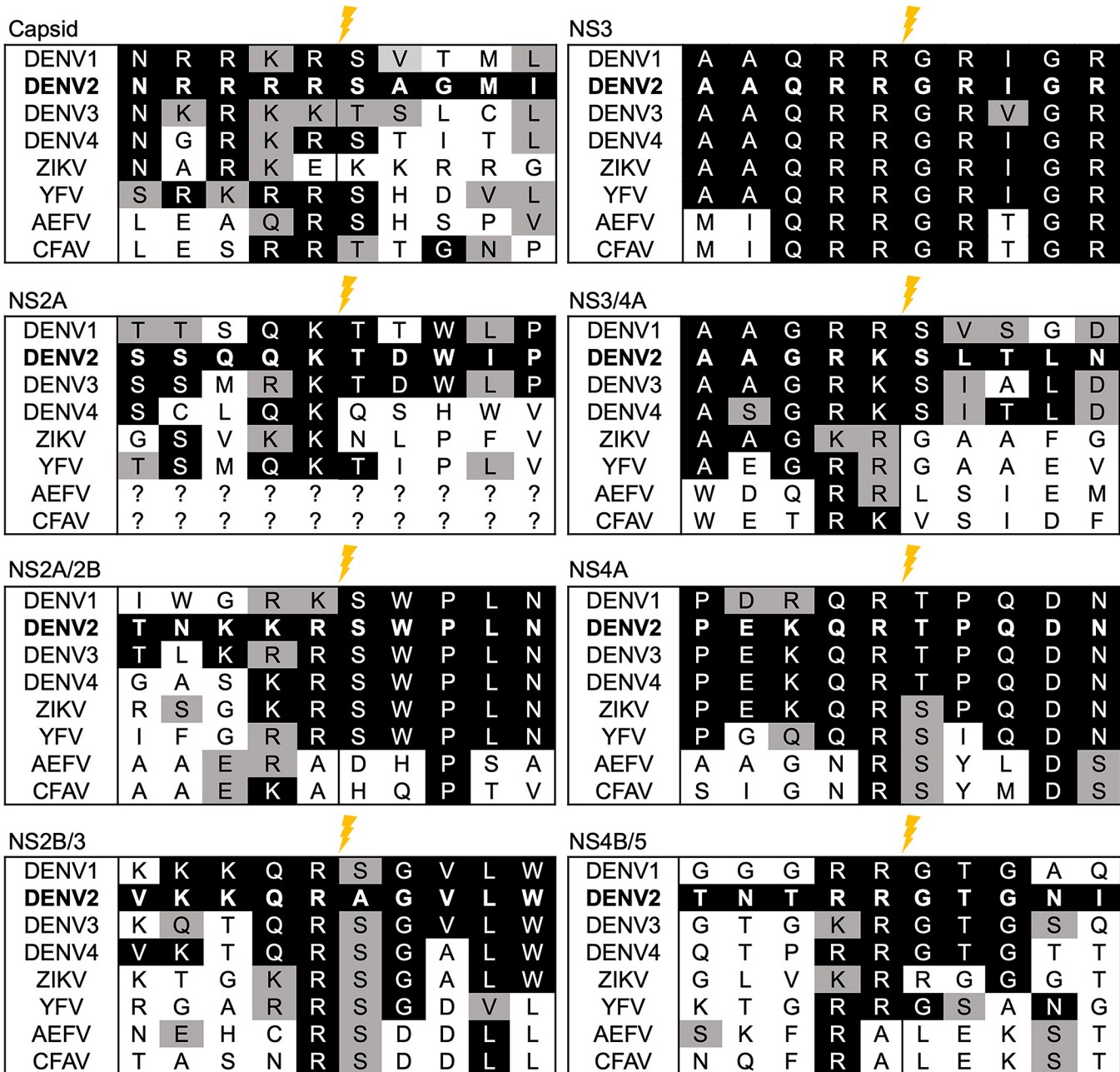

**Fig 1. Alignment of NS3 cleavage sites within the flavivirus genus.** Sequence alignment of NS3 cleavage sites across six medically significant and two insect-specific flaviviruses. Black and grey highlights represent identical and similar residues, respectively. The lightning bolt indicates the NS2B3 protease cleavage site.

measurable reversal of the quenching phenotype that would be proportionate to the amount of cleavage occurring (Fig 2). We generated nine reporters, eight with linkers encoding DENV2-- derived cleavage sites and one with EGFP fused directly to the QP (collectively referred to hereafter as EGFP$^Q$), along with an independent plasmid expressing the active DENV2-NS2B3 protease complex (CDENV2) (Fig 2), with successful expression of the DENV2 protease complex after transfection of the CDENV2 construct (Fig 3A).

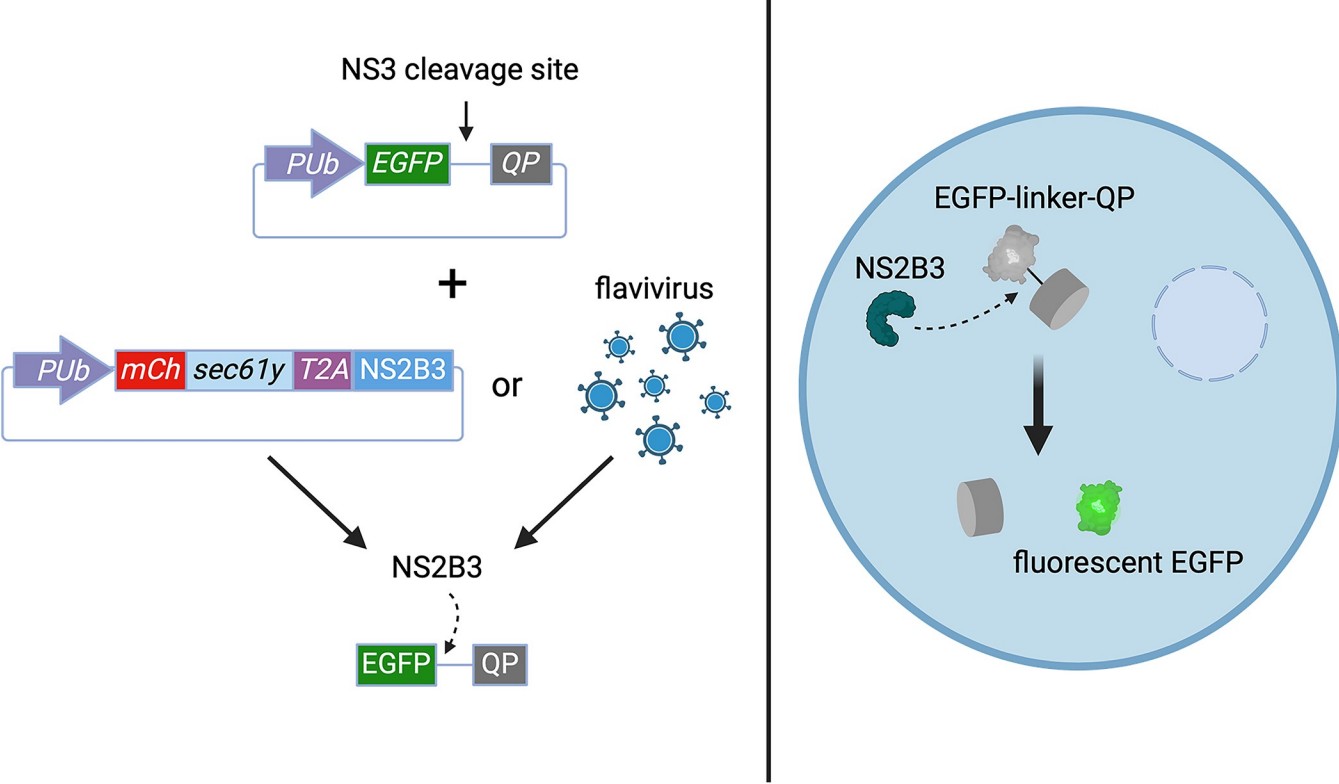

**Fig 2. Concept of the quenched-EGFP reporter.** Schematic representation of the quenched-EGFP reporter with a hydrophobic quenching peptide (QP) fused to the C-terminus of EGFP via a linker encoding a DENV2-derived NS3 cleavage site. NS2B-NS3 (NS2B3) is introduced by co-transfection of both a reporter and protease-expressing plasmid or by infecting reporter-transfected cells with a flavivirus. Illustrations were created using Biorender.com through a license to Texas A&M University. PUb–Polyubiquitin promoter sequence; mCh–mCherry red fluorescent protein; Sec61γ – γ subunit of Sec61 translocon; T2A – skipping peptide allowing for independent translation.

To determine whether the test constructs quenched EGFP fluorescence as predicted, we first transfected A20 cells with each reporter or with CDENV2 and examined the expression of each via western blot and imaging. Western blot analysis of cell lysates revealed that the A20 cells expressed similar levels of EGFP for all of the reporters (Fig 3B). However, imaging revealed that the quenching peptide did indeed suppress the fluorescence of EGFP (Fig 3C), confirming the activity of all reporters.

## CDENV2 derived DENV2-NS2B3 cleaves EGFP$^Q$ reporters *in trans*

To determine if DENV2-NS2B3 demonstrated activity against our EGFP$^Q$ reporters, A20 cells were transfected with the EGFP$^Q$ control construct alone or were co-transfected with one of the eight reporters and CDENV2 or CDENV2-S135A, an inactive protease generated by mutagenizing the catalytic serine. Fluorescence imaging of the cells 96 hours post-transfection showed highly significant increases in fluorescence intensity for six out of eight reporters when the active DENV2 protease was expressed (Fig 3D), with only reporters comprising the SQQKTDWIP and PEKQRTPQDN sites failing to show an increase in fluorescence. Correspondingly, western blot analysis on lysates of similarly transfected cells showed cleaved reporter products for the same six reporters only when the active DENV2 protease was expressed (Fig 3D). Once again, the SQQKTDWIP and PEKQRTPQDN reporters were not

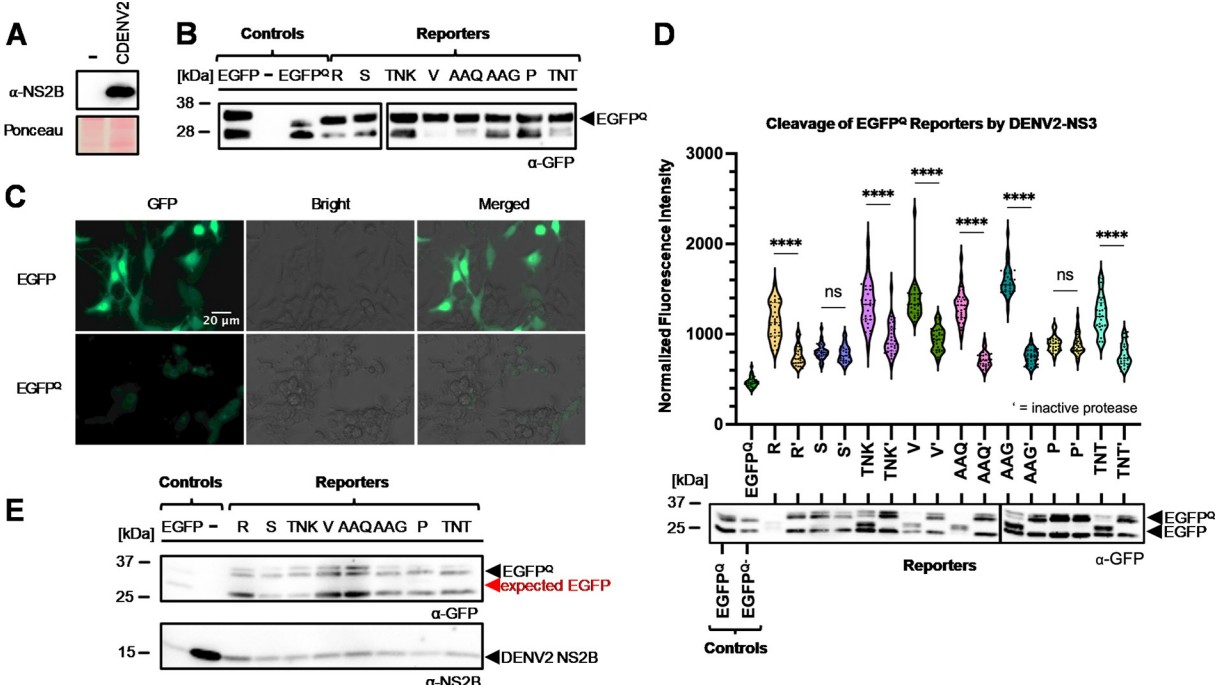

**Fig 3. Analyzing DENV2-NS2B3 activity in mosquito cells with EGFP^Q reporters.** (**A**) Western blot following transfection of A20 cells with CDENV; (-) indicates untransfected cells. (**B**) Western blot showing levels of EGFP expression of PUb-EGFP, PUb-EGFP^Q, and eight reporters. (**C**) *Ae. aegypti* A20 cells expressing EGFP and EGFP^Q imaged 48 hours post-transfection at 40x magnification under an EGFP filter or brightfield (Bright). (**D**) Co-transfected A20 cells with one of the eight reporters and the active or inactive version of the protease. Cells were imaged 96 hours post-transfection at 40x magnification (n = 30). The results are the average of three experiments (****: P ≤ 0.0001). Differences between the fluorescence intensity of the inactive-protease expressing groups and active-protease expressing groups were analyzed statistically using Mann-Whitney or independent t-tests. Cell lysates were harvested 96 hours post-transfection followed by western blotting. The cleaved (EGFP) and uncleaved (EGFP^Q) reporters are indicated by the black arrow. The reporters are represented by the first or first three letters of each NS3 cleavage site. EGFPQ–EGFP with quenching peptide and no linker; EGFPQ-–EGFP with quenching peptide and no linker co-expressed with active protease. (**E**) Western blot of A20 cells expressing EGFP^Q and the eight reporters six days following infection with DENV2 at an MOI of 0.01, with infection performed 24 hours post-transfection. Blots were probed for EGFP (α-EGFP) or DENV NS2B (α-NS2B).

cleaved. These results demonstrate that majority of the reporters respond to DENV2-NS2B3 cleavage as intended.

## DENV2-NS2B3 protease does not process EGFP^Q reporters during infection

We next wanted to determine if our EGFP^Q reporters were cleaved by NS2B3 protease during viral infection. A20 cells expressing reporter or control constructs were infected with DENV2 at 24 hours post-transfection at an MOI of 0.01. Western blot analysis failed to reveal evidence of cleavage of any of the reporters, despite the presence of DENV2-NS2B (Fig 3E). This result suggests that the DENV2 protease either could not access our cytoplasmic reporters, or that the protease expression level was insufficient to perform both cleavage of the viral polyproteins as well as *in trans* cleavage of the reporters.

## EGFP^Q reporters encoding DENV2-derived cleavage sites are processed by other flavivirus proteases

We selected four (RRRRSAG, TNKKRSWPLN, AAQRRGRIGR, and AAGRKSLTLN) of the six sites exhibiting trans-mediated cleavage by NS2B3 for further analysis based on overall

activity and level of conservation across flaviviruses. In order to determine whether these DENV2-derived sites could be cleaved by other flavivirus proteases, we cloned genetic fragments encoding NS2B3 proteases from six other medically significant flaviviruses (DENV1, DENV3-4, ZIKV, and YFV) and two insect-specific flaviviruses [Aedes flavivirus (AEFV) and cell-fusing agent virus (CFAV)] into our original effector in place of the DENV2 protease. A20 cells were transfected with each of the four EGFP$^Q$ reporters alone or co-transfected with an effector encoding an exogenous protease. Western blot analysis showed cleavage of various degrees for all four reporters by all of the medically significant flavivirus proteases, as evidenced by the appearance of a fragment corresponding to the size expected for EGFP without the quenching peptide (Fig 4). All reporters, except for RRRRSAG, were also cleaved to some extent by the insect-specific flavivirus proteases. Live imaging of the cells yielded more variable outcomes, with only some protease/cleavage sites combinations resulting in a significant increase in fluorescence (S1 Fig). We interpret this as a consequence of decreased sensitivity of

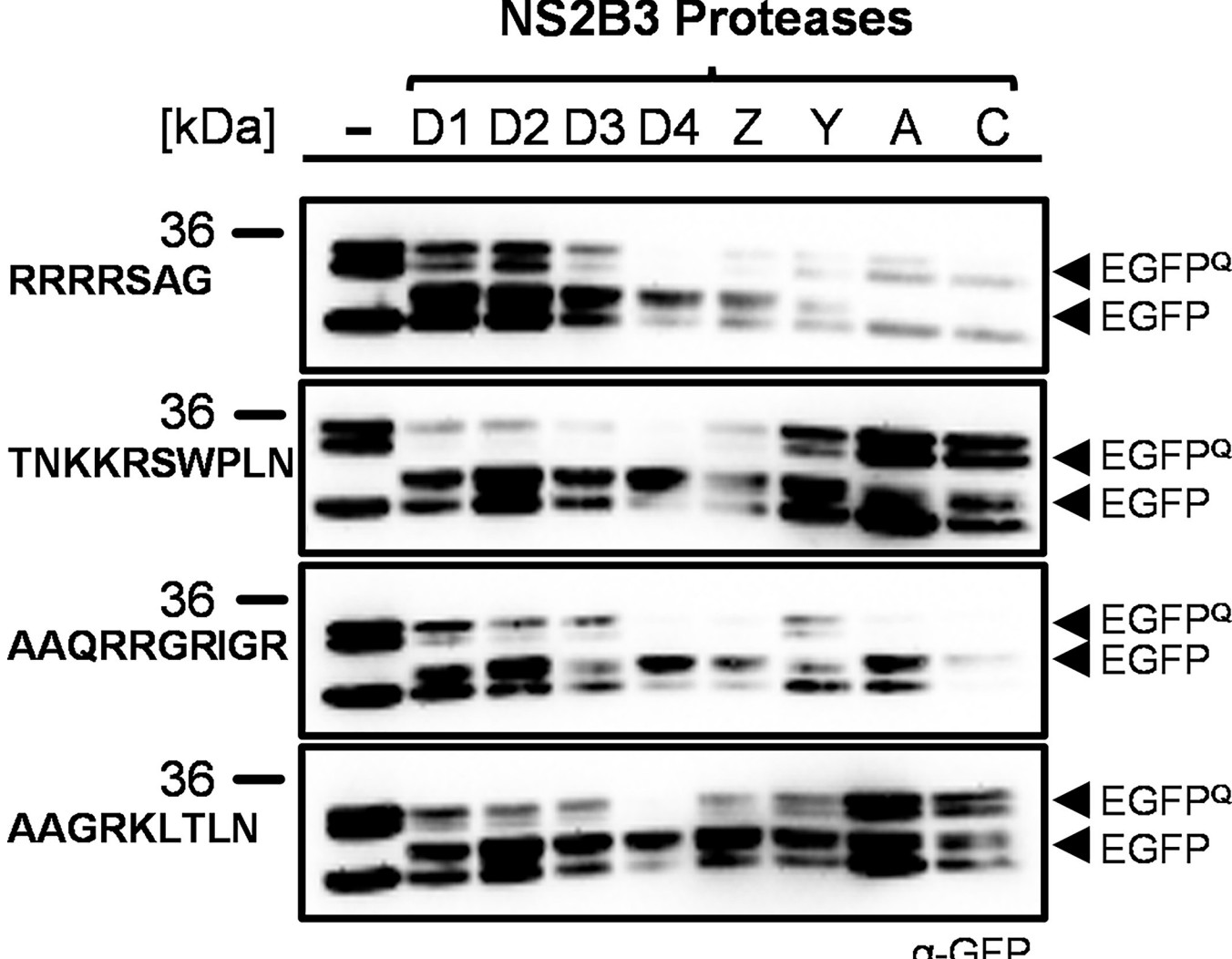

**Fig 4. Cleavage of EGFP$^Q$ reporters by exogenous flaviviral NS2B3 proteases.** Western blot of lysates obtained 96 hours post-transfection of A20 cells with reporters alone (-) or co-transfected with the indicated flavivirus protease [D1, DENV1; D2, DENV-2; D3, DENV-3; D4, DENV-4; Z, ZIKV; Y, YFV; A, AEFV; C, CFAV]. Images were derived from different exposures.

the quenched reporter assay in comparison with Western analysis, where partial cleavage events are more readily visible. Together, these results confirm that the specific cleavage sites derived from the DENV2 virus genome can be effectively processed *in trans* by a wide range of flavivirus proteases in mosquito cells, provided the protease is sufficiently expressed.

## NS3 consensus sequence is processed by a number of flavivirus proteases

While this work was in progress, a consensus motif (AEAAKRRSAGLNEM) was reported to be cleaved by DENV, ZIKV, and YFV [39]. In order to determine if this consensus site would be more efficacious as a linker than our previously tested cleavage sites, we generated an additional EGFP$^Q$ reporter. A20 cells were transfected with the new reporter alone or co-transfected with effectors encoding an exogenous protease (Fig 5). Fluorescence imaging of the cells 96 hours post-transfection showed significant increases in fluorescence intensity when the reporter was co-expressed with DENV2 and DENV4 NS2B3 proteases (Fig 5). Western blot

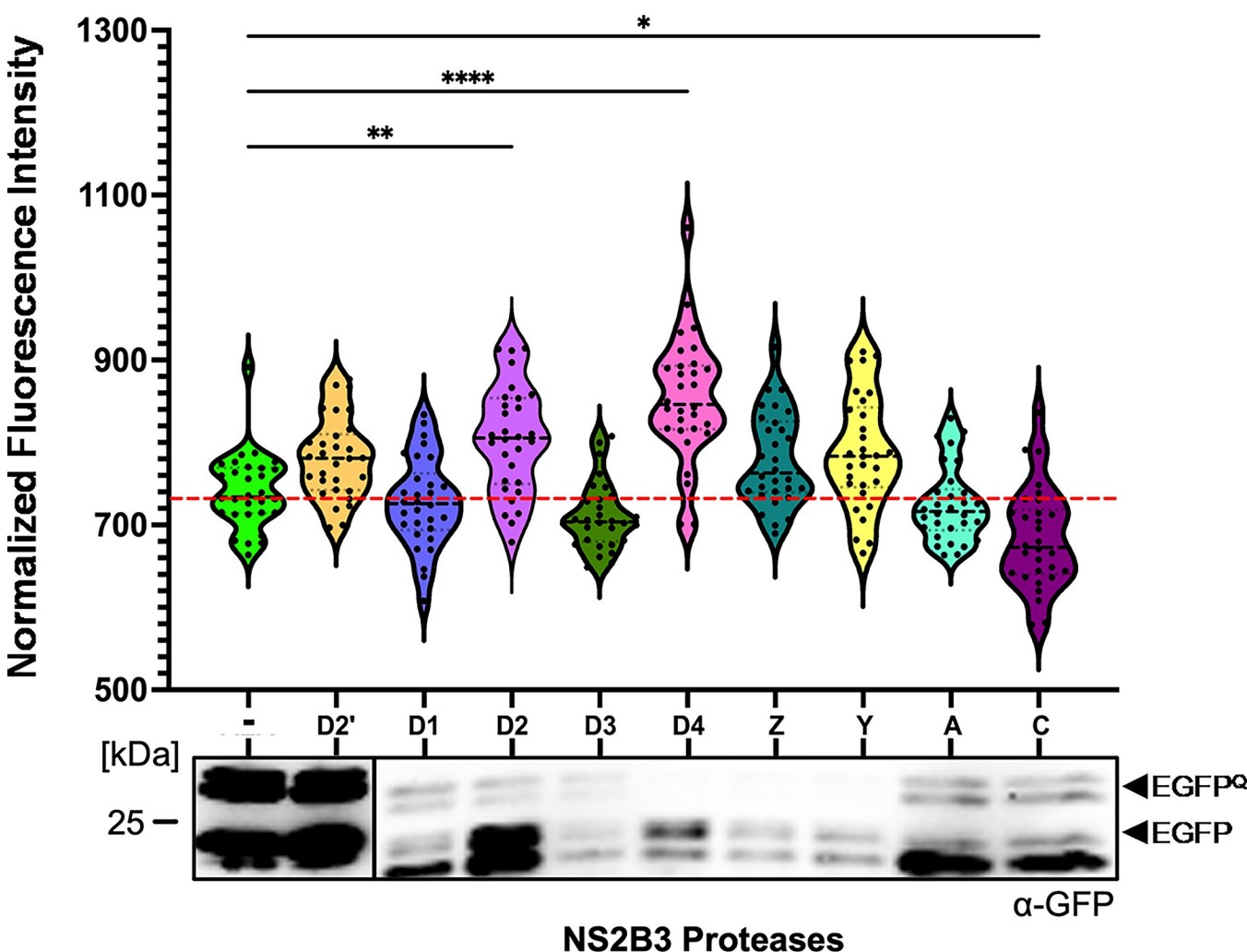

**Fig 5. Cleavage of AEAAKRRSAGLNEM encoding reporter by exogenous flaviviral NS2B3 proteases.** Cells were imaged 96 hours post-transfection at 40x magnification (n = 30). The results are the average of three experiments (*: ≤ 0.05, **: ≤ 0.01, ****: P ≤ 0.0001). Differences amongst the groups were analyzed statistically using a Kruskal-Wallis test, red horizontal line indicates the mean of the no-protease control sample. D2', inactive DENV2 NS2B3; D1, DENV1; D2, DENV-2; D3, DENV-3; D4, DENV-4; Z, ZIKV; Y, YFV; A, AEFV; C, CFAV. Cell lysates were harvested 96 hours post-transfection followed by western blotting, EGFP$^Q$ and cleaved EGFP are indicated by black arrowheads.

analysis on the cell lysates of similarly transfected cells showed that both the medically signifi-cant and insect specific flavivirus proteases were able to cleave the reporter to some extent (Fig 5). These data further demonstrate that plasticity in flavivirus NS3 protease substrate specific-ity extends also to mosquito cells.

## Analyzing the dissociation of flavivirus NS2B3 proteases from EGFP$^{Q}$ reporters

As we observed that in some instances the Western blot data and fluorescence data were con-tradictory (for example, both DENV4 and CFAV proteases appeared to cleave the entirety of the expressed EGFP$^{Q}$ reporter in the western blot, but the imaging results often showed a decrease in fluorescence intensity as opposed to an increase) (Figs 4 and 5). Seeing as how the reporters had been completely cleaved, we considered the possibility that the proteases were remaining bound to the reporter after processing and hampering proper folding of the fluores-cent protein. To investigate this, A20 cells were transfected with the RRRRSAG, TNKKRSWPLN, AAQRRGRIGR, and AAGRKSLTLN reporters, as well as co-transfected with the reporters and effectors encoding DENV2 and exogenous proteases. Intracellular cross-linking with disuccinimidyl suberate (DSS) was performed prior to harvesting the cell lysates in an attempt to prevent dissociation of any protease from the EGFP substrate that might occur during subsequent lysis. However, western blot analysis of the lysates did not reveal a band around the 66 kDa range, which would be the approximate molecular weight of the reporters (~32 kDa) and proteases (~34 kDa) combined (S2 Fig). Thus, the reason for the decrease in fluorescence intensity from some of the cleaved reporters remains unresolved.

## CDENV2 derived DENV2-NS2B3 processes ER-tethered EGFP reporters

Flavivirus replication occurs inside of virus-induced ER membrane invaginations called vesicle packets [2], while the EGFP$^{Q}$ reporters we generated are expected to be completely cyto-plasmic. These vesicle packets have small pores for openings that connect the inside of the structure to the cytoplasm [2]. It is within these invaginations that the viral non-structural (NS) proteins make up the replication complex (RC) [3, 44, 45]. We postulated that the viral NS2B3 protease was unable to interact with the cytoplasmic EGFP$^{Q}$ reporters from within the vesicle packet during viral infection. To determine differences in localization between DENV2-NS2B3 protease when expressed alone versus in the context of the replicating virus, we generated two additional reporters; one with EGFP fused to sec61β (hereafter EGFP$^{β}$), and the other with EGFP fused to the transferrin transmembrane (TM) domain followed by a KDEL ER retention signal (hereafter EGFP$^{TM}$), the latter of which was used previously in a fla-vivirus infection reporter system [39]. In both cases, the AAGRKSLTLN cleavage site was used as a linker. To validate the ability of NS2B3 to cleave each reporter, A20 cells were transfected with pSLfa-PUb-HA-EGFP alone, EGFP$^{β}$ alone, EGFP$^{TM}$ alone, or were co-transfected with one of the reporters and either CDENV2 or CDENV2-S135A (Fig 6). Fluorescence imaging of the cells 96 hours post-transfection showed a perinuclear localization of EGFP in the cells expressing EGFP$^{β}$ alone, EGFP$^{TM}$ alone, and both EGFP$^{β}$ and EGFP$^{TM}$ with the non-func-tional DENV2 protease (Fig 6A). A20 cells co-expressing EGFP$^{TM}$ and functional DENV2-NS2B3 showed EGFP dispersed evenly throughout the cell, whereas the same was not true for cells co-expressing EGFP$^{β}$ and functional DENV2-NS2B3 (Fig 6A). Consistent with this, western blot analysis on the lysates of similarly transfected cells showed that only EGFP$^{TM}$ was cleaved when co-expressed with the active DENV2-NS2B3 protease. The same was not true for co-expression with the inactive protease (Fig 6B). These results imply that although the protease can process the AAGRKSLTLN cleavage site *in trans*, successful cleavage might be

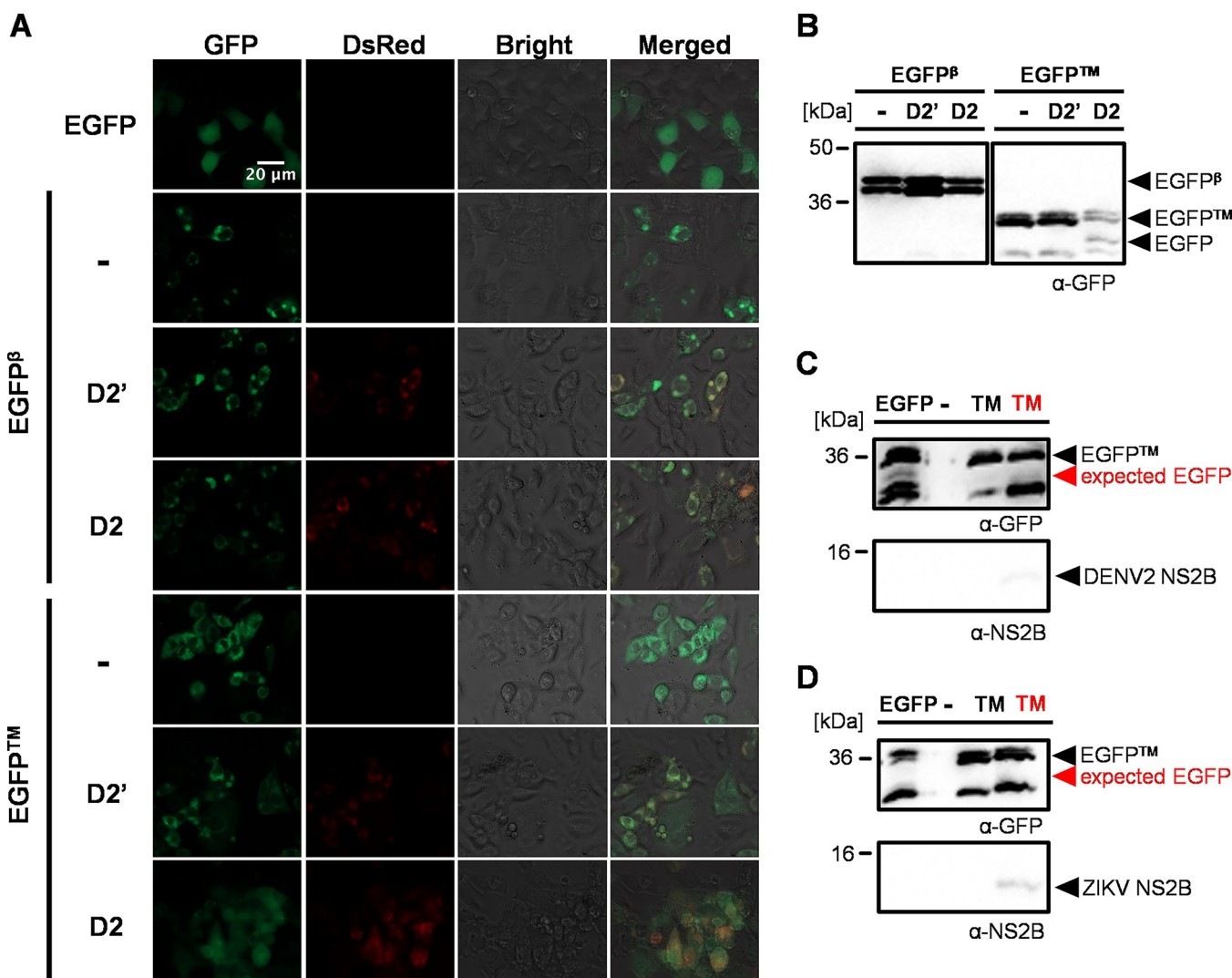

**Fig 6. Cleavage of ER-tethered EGFP reporters with DENV2-NS2B3. (A)** A20 cells transfected with expression vectors for EGFP or for ER-tethered reporters (EGFPβ or EGFPTM) alone (-), and ER-tethered reporters with inactive (D2') or active DENV2-NS2B3 (D2). Cells were imaged 96 hours post-transfection at 40x magnification. **(B)** Western blot following transfection of A20 cells with ER-tethered reporters alone (-) or with inactive (D2')/active (D2) NS2B3. Cell lysates were harvested 96 hours post-transfection. EGFPβ –EGFP-AAG-Sec61β; EGFPTM–EGFP-AAG-TM-KDEL. Images were derived from different exposures. **(C-D)** Western blot of lysates obtained 6 days following infection of EGFPTM expressing A20 cells. Cells were infected at an MOI of .01 with DENV2 **(C)** or ZIKV **(D)** 24 hours post-transfection.

hindered by the tethering technique, reinforcing the potential influence of local structure on protease access.

## Flavivirus NS2B3 proteases do not process ER-tethered EGFP reporters during infection

We next wanted to determine if our ER-tethered EGFP reporters could be processed by flavivirus proteases during viral infection. A20 cells were transfected with either pSLfa-PUb-HA-EGFP or EGFPTM alone and infected with DENV2 or ZIKV at an MOI of 0.01 24 hours post-transfection (Fig 6C and 6D). Imaging of infected EGFPTM expressing cells six days post-infection revealed perinuclear localization of EGFP (S3 Fig). Concordantly, western blot

analysis showed that the viral proteases had not cleaved the reporter, despite the presence of DENV2-NS2B3 and ZIKV-NS2B3 (Fig 6C and 6D). Afterwards we tried infecting the cells at MOIs of 0.1 and 1 (S4 Fig), but this still did not result in cleavage of the reporter. The cells also died rather quickly (~2–3 dpi) when infected at an MOI of 1, which likely precluded establishment of infection in the majority of the cells. Ultimately, these results demonstrate that even with an ER-tethered reporter, the flavivirus protease may not be able to efficiently access our reporter during infection of mosquito cells.

## Discussion

The findings from this study demonstrate the capacity of several flavivirus-derived proteases to process cleavage sites *in trans* in mosquito cells. In this study, we repurposed a cell-based reporter, originally used for monitoring flavivirus infection kinetics in mammalian cells, to assess the specificity of various flavivirus proteases to DENV2 cleavage sites in *Ae. aegypti* A20 cells [38]. We showed that proteases from six medically significant flaviviruses and two more distantly related, insect-specific flaviviruses were able to cleave three sites derived from the DENV2 polyprotein, along with an additional consensus site [39]. Ultimately, our cytoplasmic EGFP$^Q$ reporters failed to be cleaved by the protease during viral infection with DENV2. Even after tethering EGFP to the ER to bring the substrate in closer proximity to the viral protease, which resides in virus-induced vesicle packets during infection, the reporter remained uncleaved. It is possible that competition between the reporter and the viral replication machinery may strongly favor the latter, leaving insufficient NS2B3 to cleave our reporter. It may be possible to make a more competitive reporter by using TM domains and protein components from the viral genome itself, rather than from ER-localized cellular proteins. Despite this, this study has enhanced our fundamental understanding of the substrate specificity constraints of flavivirus NS3 proteases. Such insights can potentially inform the development of genetic tools meant to prevent the transmission of flaviviruses by the primary mosquito vector, *Ae. aegypti*.

Two of the three cleavage sites we had hypothesized would be cleaved by the majority of exogenous flavivirus proteases due to a number of conserved residues, specifically AAQRR-GRIGR and AAGRKSLTLN, were indeed cleaved by all the proteases we tested. Notably, AAGRKSLTLN reported the greatest fluorescence increase when cleaved by recombinant DENV2-NS2B3, followed by AAQRRGRIGR, which is consistent with a previous report that also evaluated the AAQRRGRIGR site [38]. When imaging, we did not observe the same sensitivity and consistency for reporting cleavage with any of the reporters when co-expressed with the exogenous proteases; ultimately, western blots appeared to provide more consistent readouts. It is still unclear why this was the case, as we did not find evidence that the proteases were failing to disassociate from the reporters post-cleavage and inhibiting proper folding and fluorescence of EGFP. Two cleavage sites, SSQQKTDWIP and PEKQRTPQDN, were not cleaved at all by the recombinant DENV2-NS2B3 protease. We speculate that these sites may have adopted a confirmation that the protease was unable to access and cleave. Thus, it may be possible that by modifying the linker lengths or context, cleavage of the reporters could be observed [46]. In demonstrating that various flavivirus proteases are capable of processing cleavage sites from the DENV2 genome, our study supports previous evidence of plasticity in substrate specificity of flavivirus NS3 proteases [38, 39]. Most importantly, our findings demonstrate that conceptually similar systems already developed for rendering mosquitoes refractory to a singular flavivirus may have effectiveness against a number of flaviviruses, as in the case of the study by Carvalho et al. [35].

Unexpectedly, our analysis revealed an inability of NS2B3 proteases to cleave our reporters during viral infection. This is inconsistent with the results of a previous study from which this

assay was based [38]. It is worth noting however, that previous cell-based reporter assays designed to monitor flavivirus infection and kinetics, both cytoplasmic and ER-localized, have all been carried out in mammalian cells [38, 39, 47]. It is possible that unknown differences between flavivirus replication in insect and mammalian cells impede the ability of our reporters to be cleaved as we have designed them. A transgene designed to trigger apoptosis in DENV2-infected mosquito cells utilized cleavage of the DENV2 capsid cleavage sequence, RRRRSAG, flanking both sides of tethered Michelob_x, the antagonist of the IAP [35]. However, those assays were carried out in *Ae. albopictus* C6/36 cells as opposed to *Ae. aegypti* A20 cells. The outcome we observed could very well be cell line specific, but further experiments repeating these assays in additional mosquito cell types would be needed to confirm this.

As we did not generate stably transfected cells, variable transfection efficiency between experiments could have impacted our results. In particular, the potential for limited overlap between transfected cells and infected cells could help explain the failure of reporter activation specifically during infection. While we utilized a low MOI (0.01) to limit defective interfering particle formation, higher MOIs of 0.1 and 1 were also associated with a lack of reporter cleavage. If there is indeed a limited overlap between transfected cells and infected cells, generating stably transfected cell lines for both the cytoplasmic and ER-tethered EGFP-based reporters would be appropriate for further studies.

Genetic control mechanisms have become an attractive alternative to traditional vector control methods for tempering virus transmission. This study was aimed to improve efforts focused on testing components that could be utilized in generating mosquitoes refractory to arboviruses. Previously described efforts have only been effective against individual flaviviruses or all DENV serotypes. In our study we have shown that a number of flavivirus proteases are able to process DENV2 derived cleavage sites, which suggests genetic mechanisms that result in multi-flavivirus refractoriness modulated by NS2B3 activity is a plausibility. This study can begin to potentiate the development of such mechanisms.

## Conclusion

Our study has shed light on the substrate specificity constraints of flavivirus NS3 proteases, demonstrating their capability to process cleavage sites not native to their own genome. Despite using a previously characterized cell-based reporter for monitoring flavivirus infection, we encountered unexpected challenges in detecting cleavage during viral infection. This finding underscores the complexity of flavivirus replication in mosquito cells compared to mammalian cells. Our results suggest the need for further investigation into potential cell type-specific factors during viral infection that can influence the efficacy of cell-based reporter assays. Nonetheless, our findings contribute to the ongoing efforts to develop genetic control mechanisms for preventing flavivirus transmission by insect vectors like *Ae. aegypti*. By elucidating the substrate specificity of flavivirus NS3 proteases, our study lays the groundwork for the development of genetic-based pest management strategies aimed at rendering mosquitoes refractory to multiple flaviviruses through the virus-specific activation of lethal proteins, thereby potentially mitigating the burden of flavivirus transmission.

## Supporting information

**S1 Fig. Fluorescence intensity of cells transfected individually with reporter and co-transfected with the reporter and exogenous proteases.** Cells were imaged 96 hours post-transfection at 40x magnification (n = 30). The results are the average of three experiments (*: ≤ 0.05, **: ≤ 0.01, ***: ≤ 0.001, ****: $P \leq 0.0001$). Differences amongst the groups were analyzed statistically using a Kruskal-Wallis test, red horizontal line indicates the mean of the no-protease

control sample. D1, DENV1; D2, DENV-2; D3, DENV-3; D4, DENV-4; Z, ZIKV; Y, YFV; A, AEFV; C, CFAV.
(TIF)

**S2 Fig. Intracellular crosslinking to analyze interactions between NS2B3 proteases and quenched-GFP reporters.** Western analysis of cell lysates following crosslinking (see Methods). Cells were transfected with the indicated reporter alone (-), or with the indicated protease (D2', inactive DENV2 NS2B3; D2, DENV2; D4, DENV4; C, CFAV). Images were derived from different exposures.
(TIF)

**S3 Fig. Imaging of EGFP<sup>TM</sup> expressing cells infected with DENV2 or ZIKV.** Virus-infected A20 cells expressing EGFP or EGFP<sup>TM</sup>. Cells were infected at an MOI of 0.01 with DENV2 or ZIKV 24 hours post-transfection and imaged 6 days post-infection at 40x magnification.
(TIF)

**S4 Fig. Imaging of EGFP<sup>TM</sup> expressing cells infected with DENV2.** Virus-infected A20 cells expressing EGFP or EGFP<sup>TM</sup>. Cells were infected with DENV2 at an MOI of 0.1 and 1 24 hours post-transfection. Images were derived from different exposures.
(TIF)

**S5 Fig. Raw images of western blots.** Raw, uncropped images of all Western blots used in this manuscript.
(PDF)

## Acknowledgments

We thank Kevin Myles for virus stocks. We thank Jessica Ciomperlik-Patton for support and suggestions during virus infection experiments. We also thank our collaborators, Margareth L. Capurro (University of São Paulo) and Bianca Burini (University of Florida). For reference, all raw blot images are presented in S5 Fig.

## Author Contributions

**Conceptualization:** Alexius O. Dingle, Zach N. Adelman.

**Data curation:** Alexius O. Dingle.

**Formal analysis:** Alexius O. Dingle, Zach N. Adelman.

**Funding acquisition:** Zach N. Adelman.

**Investigation:** Alexius O. Dingle, Zach N. Adelman.

**Methodology:** Alexius O. Dingle, Zach N. Adelman.

**Project administration:** Zach N. Adelman.

**Resources:** Zach N. Adelman.

**Supervision:** Zach N. Adelman.

**Validation:** Alexius O. Dingle, Zach N. Adelman.

**Visualization:** Alexius O. Dingle.

**Writing – original draft:** Alexius O. Dingle.

**Writing – review & editing:** Alexius O. Dingle, Zach N. Adelman.

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
