## [Decision Letter · Decision Letter 0]

23 Aug 2024

PONE-D-24-30069Evaluating specificity of flavivirus proteases in Aedes aegypti cells using quenched-EGFP reportersPLOS ONE

Dear Dr. Adelman,

Thank you for submitting your manuscript to PLOS ONE. After careful consideration, we feel that it has merit but does not fully meet PLOS ONE’s publication criteria as it currently stands. Therefore, we invite you to submit a revised version of the manuscript that addresses the points raised during the review process. Please submit your revised manuscript by Oct 07 2024 11:59PM. If you will need more time than this to complete your revisions, please reply to this message or contact the journal office at plosone@plos.org. Please include the following items when submitting your revised manuscript:A rebuttal letter that responds to each point raised by the academic editor and reviewer(s). You should upload this letter as a separate file labeled 'Response to Reviewers'.A marked-up copy of your manuscript that highlights changes made to the original version. You should upload this as a separate file labeled 'Revised Manuscript with Track Changes'.An unmarked version of your revised paper without tracked changes. You should upload this as a separate file labeled 'Manuscript'.

We look forward to receiving your revised manuscript.

Kind regards,

José Ramos-Castañeda, M.Sc., Ph.D

Academic Editor

PLOS ONE

**Journal Requirements:**

This project was supported by the National Institute of Allergy and Infectious Diseases (NIAID) of the National Institutes of Health (NIH) under award number R01AI149608. 

Reviewers' comments:

Reviewer's Responses to Questions

**Comments to the Author**

1. Is the manuscript technically sound, and do the data support the conclusions?

Reviewer #1: Partly

Reviewer #2: Yes

2. Has the statistical analysis been performed appropriately and rigorously? 

Reviewer #1: No

Reviewer #2: Yes

3. Have the authors made all data underlying the findings in their manuscript fully available?

Reviewer #1: Yes

Reviewer #2: Yes

4. Is the manuscript presented in an intelligible fashion and written in standard English?

Reviewer #1: Yes

Reviewer #2: Yes

5. Review Comments to the Author

**Reviewer #1:** It is not clear the statement that “although reporters remained uncleaved during viral infection, this study provides foundation for further studies aimed at developing mosquitoes that are unable to transmit most, if not all flaviviruses”

The failed EGFPQ should be further explored to explain

Authors mention “When imaging, we did not observe the same sensitivity and consistency for reporting cleavage with any of the reporters when co-expressed with the exogenous proteases….” The corelation between immunofluorescence and western blot must be further explore and discuss

Quality of images need to be improved

Although the system seems a very interesting tool it must be further explored, and new controls must be introduced to give a better explanation on the observed phenotypes and make it a robust system

Which is the difference between the 28 and the 38 KDa regarding functionality?

Figure 4 it is a central result and must be better explained

**Reviewer #2:** In this study, the authors developed a cell-based reporter in Ae. aegypti A20 cells, adapting a technique originally used for monitoring flavivirus infection kinetics in mammalian cells. This approach was employed to assess the specificity of various flavivirus proteases to DENV2 cleavage sites in Ae. aegypti A20 cells. They demonstrated that proteases from six medically significant flaviviruses, as well as two more distantly related insect-specific flaviviruses, were able to cleave three sites derived from the DENV2 polyprotein, along with an additional consensus site.

Their cytoplasmic EGFPQ reporters, however, were not cleaved by the protease during viral infection with DENV2, likely due to virus-induced vesicle packets formed during infection. It is crucial to further investigate the underlying cause of this observation.

The study also revealed that flavivirus NS3 proteases have the ability to process cleavage sites not native to their own genome. The authors suggest that their work contributes to the development of genetic control mechanisms aimed at preventing flavivirus transmission by insect vectors like Ae. aegypti. I recommend that the authors further discuss how this study could contribute to the development of such genetic control mechanisms.

Finally, I suggest increasing the resolution of the figures, as the quality of the fluorescence images is currently suboptimal.

6. PLOS authors have the option to publish the peer review history of their article (what does this mean?). If published, this will include your full peer review and any attached files.

Reviewer #1: No

Reviewer #2: No

---

## [Author Response · Author response to Decision Letter 0]

7 Oct 2024

Reviewer #1: It is not clear the statement that “although reporters remained uncleaved during viral infection, this study provides foundation for further studies aimed at developing mosquitoes that are unable to transmit most, if not all flaviviruses”

Response: This section was deleted and rephrased, as requested by the reviewer. 

The failed EGFPQ should be further explored to explain

Response: It is not clear to what this is referring to. Maybe it is related to the point below, in which case, see corresponding response. 

Authors mention “When imaging, we did not observe the same sensitivity and consistency for reporting cleavage with any of the reporters when co-expressed with the exogenous proteases….” The corelation between immunofluorescence and western blot must be further explore and discuss

Response: The reviewer does not provide a reason related to the technical conduct of our experiments as to why “the correlation between immunofluorescence and western blot must be further explored”. The EGFP-QP reporter system was developed and published by others. We adapted it to test additional cleavage sites and proteases. Our finding that in some cases this reporter system is not a reliable indicator of cleavage suggests a note of caution about relying solely on fluorescence values. Because it was unreliable, we backed up all of our experiments with Western blot data, which is less ambigious (proteins are either cleaved or not, while cleaved EGFP can be fluorescent or not depending on 3D conformation and fluorophore activation). It was not the focus of this manuscript to identify why this EGFP-QP reporter developed by others was not reliable for all constructs, nor do we make any conclusions in this regard. It seems like this is a question of IMPACT, where the reviewer feels the paper will be more IMPACTFUL if we solve this mystery, compared to if we just point out that the mystery exists. However, IMPACT is not a consideration at PLOS ONE. 

Quality of images need to be improved

Response: High-quality images were included with the submission. The images appear to be low-quality due to compression by the PLoS One submission system. In order to view the high-quality versions, reviewers need to individually download each figure rather than viewing them in the PDF. 

Although the system seems a very interesting tool it must be further explored, and new controls must be introduced to give a better explanation on the observed phenotypes and make it a robust system

Response: The main conclusions of this manuscript do not relate to the creation of a tool, despite that being part of our initial rationale. We present evidence that DENV2-based cleavage sites can be processed by a diverse array of flavivirus proteases when expressed outside of the context of an active infection. We also present evidence suggesting that such cleavage does not efficiently occur during an active infection. Our conclusions are based entirely on this evidence. The creation of a tool (ie, a system whereby toxic proteins can be produced specifically upon virus infection) will have to wait for this mystery to be resolved. Ultimately, this is a question of IMPACT. The reviewer appears to argue that developing a tool that works is more IMPACTFUL than failing to develop a tool while learning about flavivirus biology. This is irrelevant to the review criteria at PLOS ONE, where IMPACT is not a consideration. The remainder of the reviewer’s comment here cannot be productively addressed, due to a lack of information provided by the reviewer. The reviewer states that “new controls must be introduced”, but does not indicate what controls, for which experiments, and why such controls are needed beyond the controls we already included. 

Which is the difference between the 28 and the 38 KDa regarding functionality?

Response: It is not clear to what this comment is referring to. In the case of our anti-EGFP antibody, we consistently observed two different sized fragments in all samples, and we do not currently understand why. This is not a subject of investigation of this manuscript. Importantly, all blots include a negative control, so we can easily discern those fragments from new fragments that appear when treating with viral protease. 

Figure 4 it is a central result and must be better explained

Response: We have added additional information to the text here, though it would have been helpful had the reviewer indicated what specifically they found deficient in our initial explanation so our revisions could be more focused. 

Reviewer #2: In this study, the authors developed a cell-based reporter in Ae. aegypti A20 cells, adapting a technique originally used for monitoring flavivirus infection kinetics in mammalian cells. This approach was employed to assess the specificity of various flavivirus proteases to DENV2 cleavage sites in Ae. aegypti A20 cells. They demonstrated that proteases from six medically significant flaviviruses, as well as two more distantly related insect-specific flaviviruses, were able to cleave three sites derived from the DENV2 polyprotein, along with an additional consensus site.

Their cytoplasmic EGFPQ reporters, however, were not cleaved by the protease during viral infection with DENV2, likely due to virus-induced vesicle packets formed during infection. It is crucial to further investigate the underlying cause of this observation.

Response: We agree, and this is an active area of investigation. However, this comment does not appear to be related to the technical conduct of our experiments, nor whether our conclusions are supported by our data, but rather is a statement on IMPACT. That is, it appears the reviewer feels the work will be of higher impact once this mystery is solved. While we agree, IMPACT is not a criterion for publication in PLoS One. 

The study also revealed that flavivirus NS3 proteases have the ability to process cleavage sites not native to their own genome. The authors suggest that their work contributes to the development of genetic control mechanisms aimed at preventing flavivirus transmission by insect vectors like Ae. aegypti. I recommend that the authors further discuss how this study could contribute to the development of such genetic control mechanisms.

Response: We have provided further clarification of this in the Conclusions section, as suggested by the reviewer.

Finally, I suggest increasing the resolution of the figures, as the quality of the fluorescence images is currently suboptimal.

Response: High-quality images were included with the submission. The images appear to be low-quality due to compression by the PLoS One submission system. In order to view the high-quality versions, reviewers need to individually download each figure rather than viewing them in the PDF.

---

## [Decision Letter · Decision Letter 1]

6 Nov 2024

PONE-D-24-30069R1Evaluating specificity of flavivirus proteases in Aedes aegypti cells using quenched-EGFP reportersPLOS ONE

Dear Dr. Adelman,

Thank you for submitting your manuscript to PLOS ONE. After careful consideration, we feel that it has merit but does not fully meet PLOS ONE’s publication criteria as it currently stands. Therefore, we invite you to submit a revised version of the manuscript that addresses the points raised during the review process.

We look forward to receiving your revised manuscript.

Kind regards,

José Ramos-Castañeda, M.Sc., Ph.D

Academic Editor

PLOS ONE

Journal Requirements:

Additional Editor Comments:

Please heed the reviewer's suggestion regarding the title.

Reviewers' comments:

Reviewer's Responses to Questions

**Comments to the Author**

1. If the authors have adequately addressed your comments raised in a previous round of review and you feel that this manuscript is now acceptable for publication, you may indicate that here to bypass the “Comments to the Author” section, enter your conflict of interest statement in the “Confidential to Editor” section, and submit your "Accept" recommendation.

Reviewer #1: All comments have been addressed

Reviewer #2: All comments have been addressed

2. Is the manuscript technically sound, and do the data support the conclusions?

Reviewer #1: Yes

Reviewer #2: Partly

3. Has the statistical analysis been performed appropriately and rigorously? 

Reviewer #1: Yes

Reviewer #2: N/A

4. Have the authors made all data underlying the findings in their manuscript fully available?

Reviewer #1: Yes

Reviewer #2: Yes

5. Is the manuscript presented in an intelligible fashion and written in standard English?

Reviewer #1: Yes

Reviewer #2: Yes

6. Review Comments to the Author

Reviewer #1: - Review typos in the abstract “virusThese” “trans..”

- Since you are attempting to use it in transgenic refractory mosquitoes would you consider to include in your discussion

have frequent these proteolytic sites are in mosquito cells

proteins.

- For future work, how different promoters or enhancers would function for proteases activity

- If these proteases did not have effect on natural infections, which will be the way to go arount that

Reviewer #2: Given that, in certain instances, this reporter system did not reliably indicate cleavage events, the authors should consider revising the title, potentially omitting the phrase 'using quenched-EGFP reporters.' This is my final recommendation for the manuscript. Moreover, it is worth noting that this reporter system has been previously published.

7. PLOS authors have the option to publish the peer review history of their article (what does this mean?). If published, this will include your full peer review and any attached files.

Reviewer #1: **Yes: **Ma Isabel Salazar

Reviewer #2: No

---

## [Author Response · Author response to Decision Letter 1]

8 Nov 2024

Reviewer #1: -

 Review typos in the abstract “virusThese” “trans..”

Response: Corrected.

- Since you are attempting to use it in transgenic refractory mosquitoes would you consider to include in your discussion have frequent these proteolytic sites are in mosquito cells proteins.

Response: Our data indicate that in the absence of viral protease, there is no cleavage of the reporter by cellular proteins. It is possible that viral protease can cleave mosquito proteins, but this would already be occurring in a natural infection, and is not something we are attempting to control in these experiments. 

- For future work, how different promoters or enhancers would function for proteases activity

Response: The relevance of the reviewer’s comment here is not clear. Additional promoters/enhancers to increase protease expression are not needed, as we readily observed cleavage of reporters with our current constructs. Lack of activity was only associated with virus infection, but in that case the amount of protease is solely dependent on viral replication, which is not controlled by promoter/enhancers. 

- If these proteases did not have effect on natural infections, which will be the way to go arount that

Response: We added the following to the discussion “Competition between the reporter and the viral replication machinery may strongly favor the latter, leaving insufficient NS2B3 to cleave our reporter. It may be possible to make a more competetive reporter by using TM domains and protein components from the viral genome itself, rather than from ER-localized cellular proteins.”

Reviewer #2: 

Given that, in certain instances, this reporter system did not reliably indicate cleavage events, the authors should consider revising the title, potentially omitting the phrase 'using quenched-EGFP reporters.' 

Response: Title has been revised as recommended.

This is my final recommendation for the manuscript. Moreover, it is worth noting that this reporter system has been previously published.

Response: This is presented in the Introduction, lines 104-109, as well as lines 121-122. 

 “A cell-based fluorescent reporter was able to detect and monitor NS2B3 activity during flavivirus infection in live cells [38]. Flavivirus protease activity was detected by the resultant GFP fluorescence increase in cells that expressed fluorescently-quenched NS2B3 cleavable reporters [38]. Another study developed a dual-fluorescent reporter system to monitor flavivirus infection and endoplasmic reticulum (ER) manipulation [39].”

“This was achieved by adapting the previously described fluorescent activatable reporter of NS2B3 protease activity to assays in Ae. aegypti cells [38].”

---

## [Editor Report · Decision Letter 2]

12 Nov 2024

Evaluating the specificity of flavivirus proteases in Aedes aegypti cells for dengue virus 2-derived cleavage sites

PONE-D-24-30069R2

Dear Dr. Adelman,

We’re pleased to inform you that your manuscript has been judged scientifically suitable for publication and will be formally accepted for publication once it meets all outstanding technical requirements.

Kind regards,

José Ramos-Castañeda, M.Sc., Ph.D

Academic Editor

PLOS ONE
---

## [Editor Report · Acceptance letter]

19 Nov 2024

PONE-D-24-30069R2 

PLOS ONE

Dear Dr. Adelman, 

I'm pleased to inform you that your manuscript has been deemed suitable for publication in PLOS ONE. Congratulations! Your manuscript is now being handed over to our production team.

Kind regards, 

on behalf of

Dr. José Ramos-Castañeda 

Academic Editor

PLOS ONE